# DeBERTaV3: Improving DeBERTa using ELECTRA-Style Pre-Training with Gradient-Disentangled Embedding Sharing

**Pengcheng He[1], Jianfeng Gao[2], Weizhu Chen[1]**
[1] Microsoft Azure AI
[2] Microsoft Research
{penhe,jfgao,wzchen}@microsoft.com

## ABSTRACT

This paper presents a new pre-trained language model, DeBERTaV3, which improves the original DeBERTa model by replacing masked language modeling (MLM) with replaced token detection (RTD), a more sample-efficient pre-training task. Our analysis shows that vanilla embedding sharing in ELECTRA hurts training efficiency and model performance, because the training losses of the discriminator and the generator pull token embeddings in different directions, creating the "tug-of-war" dynamics. We thus propose a new *gradient-disentangled embedding sharing* method that avoids the tug-of-war dynamics, improving both training efficiency and the quality of the pre-trained model. We have pre-trained DeBERTaV3 using the same settings as DeBERTa to demonstrate its exceptional performance on a wide range of downstream natural language understanding (NLU) tasks. Taking the GLUE benchmark with eight tasks as an example, the DeBERTaV3 Large model achieves a 91.37% average score, which is 1.37% higher than DeBERTa and 1.91% higher than ELECTRA, setting a new state-of-the-art (SOTA) among the models with a similar structure. Furthermore, we have pre-trained a multilingual model mDeBERTaV3 and observed a larger improvement over strong baselines compared to English models. For example, the mDeBERTaV3 Base achieves a 79.8% zero-shot cross-lingual accuracy on XNLI and a 3.6% improvement over XLM-R Base, creating a new SOTA on this benchmark. Our models and code are publicly available at https://github.com/microsoft/DeBERTa.

## 1 INTRODUCTION

Recent advances in Pre-trained Language Models (PLMs) have created new state-of-the-art results on many natural language processing (NLP) tasks. While scaling up PLMs with billions or trillions of parameters (Raffel et al., 2020; Radford et al., 2019; Brown et al., 2020; He et al., 2020; Fedus et al., 2021) is a well-proved way to improve the capacity of the PLMs, it is more important to explore more energy-efficient approaches to build PLMs with fewer parameters and less computation cost while retaining high model capacity.

Towards this direction, there are a few works that significantly improve the efficiency of PLMs. The first is RoBERTa (Liu et al., 2019) which improves the model capacity with a larger batch size and more training data. Based on RoBERTa, DeBERTa (He et al., 2020) further improves the pre-training efficiency by incorporating disentangled attention which is an improved relative-position encoding mechanism. By scaling up to 1.5B parameters, which is about an eighth of the parameters of xxlarge T5 (Raffel et al., 2020), DeBERTa surpassed human performance on the SuperGLUE (Wang et al., 2019a) leaderboard for the first time. The second new pre-training approach to improve efficiency is Replaced Token Detection (RTD), proposed by ELECTRA (Clark et al., 2020). Unlike BERT (Devlin et al., 2019), which uses a transformer encoder to predict corrupt tokens with masked language modeling (MLM), RTD uses a generator to generate ambiguous corruptions and a discriminator to distinguish the ambiguous tokens from the original inputs, similar to Generative Adversarial

Networks (GAN). The effectiveness of RTD is also verified by several works, including CoCo-LM (Meng et al., 2021), XLM-E (Chi et al., 2021), CodeBERT(Feng et al., 2020) and SmallBenchNLP (Kanakarajan et al., 2021).

In this paper, we explore two methods of improving the efficiency of pre-training DeBERTa. Following ELECTRA-style training, we replace MLM in DeBERTa with RTD where the model is trained as a discriminator to predict whether a token in the corrupt input is either original or replaced by a generator. We show that DeBERTa trained with RTD significantly outperforms the model trained using MLM. The second is a new embedding sharing method. In ELECTRA, the discriminator and the generator share the same token embeddings. However, our analysis shows that embedding sharing hurts training efficiency and model performance, since the training losses of the discriminator and the generator pull token embeddings into opposite directions. This is because the training objectives between the generator and the discriminator are very different. The MLM used for training the generator tries to pull the tokens that are semantically similar close to each other while the RTD of the discriminator tries to discriminate semantically similar tokens and pull their embeddings as far as possible to optimize the binary classification accuracy, causing a conflict between their training objectives. In other words, this creates the "tug-of-war" dynamics that reduces the training efficiency and the model quality, as illustrated in Hadsell et al. (2020). On the other hand, we show that using separated embeddings for the generator and the discriminator results in significant performance degradation when we fine-tune the discriminator on downstream tasks, indicating the merit of embedding sharing, e.g., the embeddings of the generator are beneficial to produce a better discriminator, as argued in Clark et al. (2020). To balance these tradeoffs, we propose a new *gradient-disentangled embedding sharing* (GDES) method where the generator shares its embeddings with the discriminator but stops the gradients from the discriminator to the generator embeddings. This way, we avoid the tug-of-war effect and preserve the benefits of embedding sharing. We empirically demonstrate that GDES improves both pre-training efficiency and the quality of the pre-trained models.

We pre-train four variants of DeBERTaV3 models, i.e., DeBERTaV3$_{large}$, DeBERTaV3$_{base}$, DeBERTaV3$_{small}$ and DeBERTaV3$_{xsmall}$. We evaluate them on various representative natural language understanding (NLU) benchmarks and set new state-of-the-art numbers among models with a similar model structure. For example, DeBERTaV3$_{large}$ surpasses previous SOTA models with a similar model structure on GLUE (Wang et al., 2019b) benchmark with an average score over +1.37%, which is significant. DeBERTaV3$_{base}$ achieves a 90.6% accuracy score on the MNLI-matched (Williams et al., 2018) evaluation set and an 88.4% F1 score on the SQuAD v2.0 (Rajpurkar et al., 2018) evaluation set. This improves DeBERTa$_{base}$ by 1.8% and 2.2%, respectively. Without knowledge distillation, DeBERTaV3$_{small}$ and DeBERTaV3$_{xsmall}$ surpasses previous SOTA models with a similar model structure on both MNLI-matched and SQuAD v2.0 evaluation set by more than 1.2% in accuracy and 1.3% in F1, respectively. We also train DeBERTaV3$_{base}$ on the CC100 (Conneau et al., 2020) multilingual data using a similar setting as XLM-R (Conneau et al., 2020) but with only a third of the training passes. We denote the model as mDeBERTaV3$_{base}$. Under the cross-lingual transfer setting, mDeBERTaV3$_{base}$ achieves a 79.8% average accuracy score on the XNLI (Conneau et al., 2018) task, which outperforms XLM-R$_{base}$ and mT5$_{base}$ (Xue et al., 2021) by 3.6% and 4.4%, respectively. This makes mDeBERTaV3 the best model among multi-lingual models with a similar model structure. All these results strongly demonstrate the efficiency of DeBERTaV3 models and set a good base for future exploration towards more efficient PLMs.

## 2 Background

### 2.1 Transformer

A Transformer-based language model is composed of $L$ stacked Transformer blocks (Vaswani et al., 2017). Each block contains a multi-head self-attention layer followed by a fully connected positional feed-forward network. The standard self-attention mechanism lacks a natural way to encode word position information. Thus, existing approaches add a positional bias to each input word embedding so that each input word is represented by a vector whose value depends on both its content and position. The positional bias can be implemented using absolute position embedding (Vaswani et al., 2017; Brown et al., 2020; Devlin et al., 2019) or relative position embedding (Huang et al., 2018; Yang et al., 2019). Several studies have shown that relative position representations are more effective

for natural language understanding and generation tasks (Dai et al., 2019; Shaw et al., 2018; He et al., 2020).

## 2.2 DeBERTa

DeBERTa improves BERT with two novel components: DA (Disentangled Attention) and an enhanced mask decoder. Unlike existing approaches that use a single vector to represent both the content and the position of each input word, the DA mechanism uses two separate vectors: one for the content and the other for the position. Meanwhile, the DA mechanism's attention weights among words are computed via disentangled matrices on both their contents and relative positions. Like BERT, DeBERTa is pre-trained using masked language modeling. The DA mechanism already considers the contents and relative positions of the context words, but not the absolute positions of these words, which in many cases are crucial for the prediction. DeBERTa uses an enhanced mask decoder to improve MLM by adding absolute position information of the context words at the MLM decoding layer.

## 2.3 ELECTRA

### 2.3.1 Masked Language Model

Large-scale Transformer-based PLMs are typically pre-trained on large amounts of text to learn contextual word representations using a self-supervision objective, known as MLM (Devlin et al., 2019). Specifically, given a sequence $\boldsymbol{X} = \{x_i\}$, we corrupt it into $\tilde{\boldsymbol{X}}$ by masking 15% of its tokens at random and then train a language model parameterized by $\theta$ to reconstruct $\boldsymbol{X}$ by predicting the masked tokens $\tilde{x}$ conditioned on $\tilde{\boldsymbol{X}}$:

$$\max_{\theta} \log p_{\theta}(\boldsymbol{X}|\tilde{\boldsymbol{X}}) = \max_{\theta} \sum_{i \in \mathcal{C}} \log p_{\theta}(\tilde{x}_i = x_i|\tilde{\boldsymbol{X}}) \tag{1}$$

where $\mathcal{C}$ is the index set of the masked tokens in the sequence. The authors of BERT propose to keep 10% of the masked tokens unchanged, another 10% replaced with randomly picked tokens and the rest replaced with the `[MASK]` token.

### 2.3.2 Replaced token detection

Unlike BERT, which uses only one transformer encoder and trained with MLM, ELECTRA was trained with two transformer encoders in GAN style. One is called generator trained with MLM; the other is called discriminator trained with a token-level binary classifier. The generator is used to generate ambiguous tokens to replace masked tokens in the input sequence. Then the modified input sequence is fed to the discriminator. The binary classifier in the discriminator needs to determine if a corresponding token is either an original token or a token replaced by the generator. We use $\theta_G$ and $\theta_D$ to represent the parameters of the generator and the discriminator, respectively. The training objective in the discriminator is called RTD (Replaced Token Detection). The loss function of the generator can be written as,

$$L_{MLM} = \mathbb{E}\left(-\sum_{i \in \mathcal{C}} \log p_{\theta_G}\left(\tilde{x}_{i,G} = x_i|\tilde{\boldsymbol{X}}_G\right)\right) \tag{2}$$

, where $\tilde{\boldsymbol{X}}_G$ is the input to the generator by randomly masking 15% tokens in $\boldsymbol{X}$.

The input sequence of the discriminator is constructed by replacing masked tokens with new tokens sampled according to the output probability from the generator:

$$\tilde{x}_{i,D} = \begin{cases} \tilde{x}_i \sim p_{\theta_G}\left(\tilde{x}_{i,G} = x_i|\tilde{\boldsymbol{X}}_G\right), & i \in \mathcal{C} \\ x_i, & i \notin \mathcal{C} \end{cases} \tag{3}$$

The loss function of the discriminator is written as,

$$L_{RTD} = \mathbb{E}\left(-\sum_{i} \log p_{\theta_D}\left(\mathbb{1}\left(\tilde{x}_{i,D} = x_i\right)|\tilde{\boldsymbol{X}}_D, i\right)\right) \tag{4}$$

, where $\mathbb{1}(\cdot)$ is the indicator function and $\tilde{\boldsymbol{X}}_D$ is the input to the discriminator constructed via equation 3. In ELECTRA, $L_{MLM}$ and $L_{RTD}$ are optimized jointly, $L = L_{MLM} + \lambda L_{RTD}$, where $\lambda$ is the weight of the discriminator loss $L_{RTD}$, which was set to 50 in ELECTRA.

## 3 DEBERTAV3

This section describes DeBERTaV3, which improves DeBERTa by using the RTD training loss of Clark et al. (2020) and a new weight-sharing method.

### 3.1 DEBERTA WITH RTD

Since RTD in ELECTRA and the disentangled attention mechanism in DeBERTa have proven to be sample-efficient for pre-training, we propose a new version of DeBERTa, referred to as *DeBERTaV3*, by replacing the MLM objective used in DeBERTa with the RTD objective to combine the strengths of the latter.

In this implementation, Wikipedia and the bookcorpus (Zhu et al., 2015) are used as training data, following the base model configuration of Devlin et al. (2019). The generator is the same width as the discriminator but is half the depth. The batch size is set to 2048, and the model is trained for 125,000 steps with a learning rate of 5e-4 and warmup steps of 10,000. Following Clark et al. (2020), we use $\lambda = 50$ with the same optimization hyperparameters. We validate the effectiveness of DeBERTaV3 on two representative NLU tasks, i.e., MNLI and SQuAD v2.0. The results, presented in ① of Table 2, show that DeBERTaV3 significantly outperforms DeBERTa, i.e., +2.5% on the MNLI-m accuracy and +3.8% on the SQuAD v2.0 F1.

In the next two subsections, we will show that the performance of DeBERTaV3 can be further improved by replacing token Embedding Sharing (ES) used for RTD, originally proposed in Clark et al. (2020), by a new Gradient-Disentangled Embedding Sharing (GDES) method. We start with an analysis of ES in Section 3.2.

### 3.2 TOKEN EMBEDDING SHARING IN ELECTRA

To pre-train ELECTRA, we use a generator and a discriminator that share token embeddings, as shown in Figure 1 (a). This method, called Embedding Sharing (ES), allows the generator to provide informative inputs for the discriminator and reduces the number of parameters to learn. However, it also creates a multitask learning problem, where the generator's and the discriminator's objectives interfere with each other and slow down the training convergence.

Let $\boldsymbol{E}$ and $g_{\boldsymbol{E}}$ be the token embeddings and their gradients, respectively. In each training step of ELECTRA, we compute $g_{\boldsymbol{E}}$ by back-propagating the errors from both the generator's Masked Language Modeling (MLM) loss and the discriminator's Replaced Token Detection (RTD) loss, as $g_{\boldsymbol{E}} = \frac{\partial L_{MLM}}{\partial \boldsymbol{E}} + \lambda \frac{\partial L_{RTD}}{\partial \boldsymbol{E}}$. This equation means that the token embeddings are updated by balancing the gradients from the two tasks, which can be seen as a tug-of-war procedure (Hadsell et al., 2020). The procedure can eventually converge if we control the update speed carefully (e.g., using a small learning rate or gradient clipping), but it can be very inefficient if the two tasks have different optimal directions for the embeddings. This is indeed the case for MLM and RTD, because they have opposite effects on the token embeddings. MLM encourages the embeddings of semantically similar tokens to be close to each other, while RTD tries to separate them to make the classification easier.

To verify our hypothesis, we implement a variant of ELECTRA that does not share token embeddings between the generator and the discriminator, which we refer to as No Embedding Sharing (NES), as illustrated in Figure 1 (b). In NES, we update the generator and the discriminator alternately in each training step. First, we run the generator to generate inputs for the discriminator, and then we update the generator's parameters, including its token embeddings $\boldsymbol{E}_G$, by back-propagating the MLM loss. Next, we run the discriminator using the inputs from the generator and update the discriminator's parameters, including its token embeddings $\boldsymbol{E}_D$, by back-propagating the RTD loss.

We compare ES and NES on three aspects: convergence speed, quality of the token embeddings, and performance on downstream Natural Language Understanding (NLU) tasks. Figure 2 shows that NES converges faster than ES, as expected, because it avoids the conflicting gradients between the two

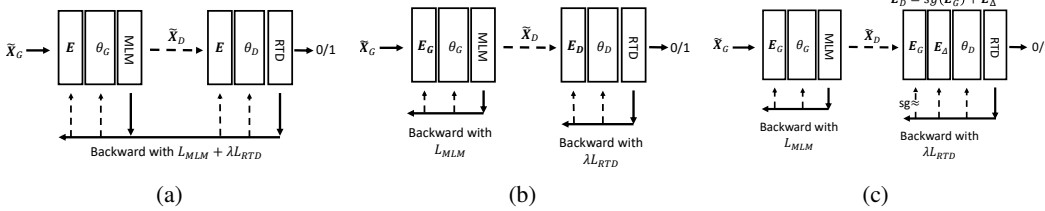

(a)  (b)  (c)

Figure 1: Illustration of different embedding sharing methods. (a) ES: $\boldsymbol{E}$, $\theta_G$ and $\theta_D$ will be jointly updated in a single backward pass with regards to $L_{MLM} + \lambda L_{RTD}$ . (b) NES: $\boldsymbol{E}_G$ and $\theta_G$ will first be updated via the backward pass with regards to $L_{MLM}$, then $\boldsymbol{E}_D$ and $\theta_D$ will be updated via the backward pass with regards to $\lambda L_{RTD}$ . (c) GDES: $\boldsymbol{E}_G$ and $\theta_G$ will first be updated in the backward pass with regards to $L_{MLM}$, then $\boldsymbol{E}_\Delta$ and $\theta_D$ will be updated via the backward pass with regards to $\lambda L_{RTD}$ and $\boldsymbol{E}_G$ . $sg$ is the stop gradient operator that prevents the discriminator from updating $\boldsymbol{E}_G$

Table 1: Average cosine similarity of word embeddings of the generator and the discriminator with different embedding sharing methods.

| Word Embedding Sharing | $\boldsymbol{E}_G$ | $\boldsymbol{E}_D$ | $\boldsymbol{E}_\Delta$ |
|---|---|---|---|
| ① ES | 0.02 | 0.02 | - |
| ② NES | 0.45 | 0.02 | - |
| ③ GDES | 0.45 | 0.29 | 0.02 |

Table 2: Fine-tuning results on MNLI and SQuAD v2.0 tasks of base models trained with different embedding sharing methods.

| Model | MNLI-m/mm Acc | SQuAD v2.0 F1/EM |
|---|---|---|
| BERT$_{base}$ | 84.3/84.7 | 76.3/73.7 |
| ELECTRA$_{base}$ | 85.8/- | -/- |
| DeBERTa$_{base}$ | 86.3/86.2 | 82.5/79.3 |
| DeBERTa+RTD$_{base}$ | | |
| ① ES | 88.8/88.4 | 86.3/83.5 |
| ② NES | 88.3/87.9 | 85.3/82.7 |
| ③ GDES | **89.3/89.0** | **87.2/84.5** |

downstream tasks. We then measure the average cosine similarity scores of the token embeddings [1]. Table 1 shows that NES produces two distinct embedding models, with $\boldsymbol{E}_G$ being more semantically coherent than $\boldsymbol{E}_D$. This confirms our hypothesis. However, the embeddings learned by NES do not lead to any significant improvement on two representative downstream NLU tasks (i.e., MNLI and SQuAD v2.0), as shown in Table 2. This result supports the argument of Clark et al. (2020) that ES has the advantage of making the discriminator benefit from the generator's embeddings, in addition to saving parameters. In the next subsection, we propose a new embedding sharing method that combines the strengths of ES and NES, while avoiding their drawbacks.

### 3.3 GRADIENT-DISENTANGLED EMBEDDING SHARING

We propose the Gradient-Disentangled Embedding Sharing (GDES) method to overcome the drawbacks of ES and NES while retaining their advantages. As shown in Figure 1 (c), GDES shares the token embeddings between the generator and the discriminator, which enables the two models to learn from the same vocabulary and leverage the rich semantic information encoded in the embeddings. However, unlike ES, GDES does not allow the RTD loss to affect the gradients of the generator, thus avoiding the interference and inefficiency caused by the conflicting objectives. Instead, GDES only updates the generator embeddings with the MLM loss, which ensures the consistency and coherence of the generator output. As a result, GDES can achieve the same converging speed as NES, but without sacrificing the quality of the embeddings.

To implement GDES, we re-parameterize the discriminator embeddings as $\boldsymbol{E}_D = sg(\boldsymbol{E}_G) + \boldsymbol{E}_\Delta$, where the stop gradient operator $sg$ prevents the gradients from flowing through the generator embeddings $\boldsymbol{E}_G$ and only updates the residual embeddings $\boldsymbol{E}_\Delta$. We initialize $\boldsymbol{E}_\Delta$ as a zero matrix and train the model following the NES procedure. In each iteration, we first generate the inputs for the discriminator using the generator and update both $\boldsymbol{E}_G$ and $\boldsymbol{E}_D$ with the MLM loss. Then, we run the discriminator on the generated inputs and update $\boldsymbol{E}_D$ with the RTD loss, but only through $\boldsymbol{E}_\Delta$. After training, we add $\boldsymbol{E}_\Delta$ to $\boldsymbol{E}_G$ and save the resulting matrix as $\boldsymbol{E}_D$ for the discriminator.

---

[1]We sample 10% of the word pieces from the vocabulary and calculate the average cosine similarity of every pair of two word pieces.

We conduct extensive experiments to evaluate the effectiveness of GDES compared to ES and NES. We measure the converging speed, the quality of the token embeddings, and the performance on downstream tasks. The results in Figure 2, Tables 1 and 2 demonstrate that GDES outperforms both ES and NES in various aspects. First, Figure 2 reveals that GDES converges faster than ES and matches the efficiency of NES. As GDES, ES and NES only differ at the way of embedding sharing, the computation cost of each step is the same. Second, Table 1 indicates that GDES produces two distinctive token embedding matrices, with the generator embeddings having higher average similarity

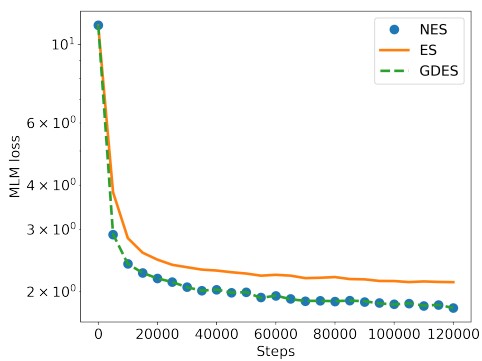

Figure 2: MLM training loss of the generator with different word embedding sharing methods.

scores than the discriminator embeddings. However, the difference is smaller than that in NES, suggesting that GDES preserves more semantic information in the discriminator embeddings through the partial weight sharing. Third, Table 2 shows that after fine-tuning, the model pre-trained with GDES achieves the best performance on two downstream tasks, MNLI and SQuAD v2.0. These results confirm that GDES is an effective weight-sharing method for language model pre-trained with MLM and RTD.

## 4 EXPERIMENT

### 4.1 MAIN RESULTS ON NLU TASKS

To further verify the effectiveness of those technologies, we combine RTD, GDES and DA (Disentangled Attention) to train models of different sizes(i.e., large, base and small) using standard pre-training settings. Since all of our experiments are modified based on DeBERTa code base and follow most of the settings of DeBERTa, we denote the new models as DeBERTaV3$_{large}$, DeBERTaV3$_{base}$, and DeBERTaV3$_{small}$. The discriminator part of DeBERTaV3$_{large}$ and DeBERTaV3$_{base}$ are the same as DeBERTa$_{large}$ and DeBERTa$_{base}$, respectively. The discriminator of DeBERTaV3$_{small}$ has the same width and attention heads as DeBERTa$_{base}$ and half the depth of DeBERTa$_{base}$, i.e., 6 layers with 768 hidden size and 12 attention heads. The generator of DeBERTaV3 has the same width as the discriminator and half the depth of the discriminator. We train those models with 160GB data, which is the same as DeBERTaV2 and RoBERTa, and use the same SentencePiece (Kudo, 2018; Sennrich et al., 2016) vocabulary as DeBERTaV2 (He et al., 2020) which contains 128,000 tokens. All the models are trained for 500,000 steps with a batch size of 8192 and warming up steps of 10,000. The learning rate for base and small model is 5e-4, while the learning rate for large model is 3e-4. Following the DeBERTa setting, we use the AdamW (Loshchilov & Hutter, 2018) optimizer which is a fixed version of Adam (Kingma & Ba, 2014) with weight decay, and set $\beta_1 = 0.9$, $\beta_2 = 0.98$ for the optimizer. After pre-training, the discriminators of those models are used for downstream task fine-tuning following the same paradigm as Transformer PLMs, such as BERT, RoBERTa, ELECTRA, and DeBERTa. We provide more details on the hyper parameters of pre-training and fine-tuning in the Appendix.

### 4.1.1 PERFORMANCE ON LARGE MODELS

Following previous studies on PLMs, we first evaluate our model on the eight NLU tasks in GLUE (Wang et al., 2019b), which are the most representative sentence classification tasks. We fine tune the pre-trained models on those tasks by plugging a classification head on top of the hidden states of the [CLS] token at the last layer. We summarize the results in Table 3, where DeBERTaV3 is compared with previous Transformer-based PLMs of similar structures (i.e., 24 layers with hidden size of 1024), including BERT, RoBERTa, XLNet (Yang et al., 2019), ALBERT (Lan et al., 2019), ELECTRA and DeBERTa. Compared to previous SOTA models, DeBERTaV3 performs consistently comparable or mostly better across all the tasks. Meanwhile, DeBERTaV3 outperforms XLNet in seven out of eight tasks. In terms of average GLUE score, DeBERTaV3 outperforms other SOTA PLMs with a large margin (> 1.3%). Particularly, compared with previous best numbers, there are big

jumps on the performance of low resource tasks (i.e., RTE (+4.4%), CoLA (+4.8%)). This indicates DeBERTaV3 is more data efficient and has a better generalization performance. We also note that the improvements on SST-2, STS-B, and MRPC are relatively small ($< 0.3\%$). We conjecture this is due to the performance on those tasks is close to be saturated, and thus even small but consistent improvements on them are valuable.

Table 3: Comparison results on the GLUE development set.

| **Model** | CoLA | QQP | MNLI-m/mm | SST-2 | STS-B | QNLI | RTE | MRPC | Avg. |
| | Mcc | Acc | Acc | Acc | Corr | Acc | Acc | Acc | |
| #Train | 8.5k | 364k | 393k | 67k | 7k | 108k | 2.5k | 3.7k | |
| BERT$_{large}$ | 60.6 | 91.3 | 86.6/- | 93.2 | 90.0 | 92.3 | 70.4 | 88.0 | 84.05 |
| RoBERTa$_{large}$ | 68.0 | 92.2 | 90.2/90.2 | 96.4 | 92.4 | 93.9 | 86.6 | 90.9 | 88.82 |
| XLNet$_{large}$ | 69.0 | 92.3 | 90.8/90.8 | **97.0** | 92.5 | 94.9 | 85.9 | 90.8 | 89.15 |
| ELECTRA$_{large}$ | 69.1 | 92.4 | 90.9/- | 96.9 | 92.6 | 95.0 | 88.0 | 90.8 | 89.46 |
| DeBERTa$_{large}$ | 70.5 | 92.3 | 91.1/91.1 | 96.8 | 92.8 | 95.3 | 88.3 | 91.9 | 90.00 |
| DeBERTaV3$_{large}$ | **75.3** | **93.0** | **91.8/91.9** | 96.9 | **93.0** | **96.0** | **92.7** | **92.2** | **91.37** |

To further evaluate the model performance, in addition to GLUE, DeBERTaV3$_{large}$ is evaluated on three categories of representative NLU benchmarks: (1) Question Answering: SQuAD v2.0, RACE (Lai et al., 2017), ReCoRD (Zhang et al., 2018), and SWAG (Zellers et al., 2018); (2) Natural Language Inference: MNLI; and (3) NER: CoNLL-2003 (Sang & De Meulder, 2003). Among those tasks, RACE, SWAG, and MNLI are fine-tuned using a same way as sentence classification tasks. SQuAD v2.0, ReCoRD and NER are fine-tuned as sequence tagging tasks, where a token classification head is plugged on top of the hidden states of each token at the last layer. For comparison, we include ALBERT$_{xxlarge}$ [2], DeBERTa$_{large}$, DeBERTa$_{1.5B}$, and Megatron (Shoeybi et al., 2019) with three different model sizes, denoted as Megatron$_{336M}$, Megatron$_{1.3B}$ and Megatron$_{3.9B}$, which are trained using the same dataset as RoBERTa. Note that Megatron$_{336M}$ has a similar model size as other models mentioned above[3]. We summarize the results in Table 4. Compared to the previous

Table 4: Results on MNLI in/out-domain, SQuAD v2.0, RACE, ReCoRD, SWAG, CoNLL 2003 NER development set. Note that missing results in literature are signified by "-".

| **Model** | MNLI-m/mm | SQuAD v2.0 | RACE | ReCoRD | SWAG | NER |
| | Acc | F1/EM | Acc | F1/EM | Acc | F1 |
| BERT$_{large}$ | 86.6/- | 81.8/79.0 | 72.0 | - | 86.6 | 92.8 |
| ALBERT$_{large}$ | 86.5/- | 84.9/81.8 | 75.2 | - | - | - |
| RoBERTa$_{large}$ | 90.2/90.2 | 89.4/86.5 | 83.2 | 90.6/90.0 | 89.9 | 93.4 |
| XLNet$_{large}$ | 90.8/90.8 | 90.6/87.9 | 85.4 | - | - | - |
| ELECTRA$_{large}$ | 90.9/- | -/88.1 | - | - | - | - |
| Megatron$_{336M}$ | 89.7/90.0 | 88.1/84.8 | 83.0 | - | - | - |
| DeBERTa$_{large}$ | 91.1/91.1 | 90.7/88.0 | 86.8 | 91.4/91.0 | 90.8 | 93.8 |
| DeBERTaV3$_{large}$ | **91.8/91.9** | **91.5/89.0** | **89.2** | **92.3/91.8** | **93.4** | **93.9** |
| ALBERT$_{xxlarge}$ | 90.8/- | 90.2/87.4 | 86.5 | - | - | - |
| Megatron$_{1.3B}$ | 90.9/91.0 | 90.2/87.1 | 87.3 | - | - | - |
| Megatron$_{3.9B}$ | 91.4/91.4 | 91.2/88.5 | 89.5 | - | - | - |
| DeBERTa$_{1.5B}$ | 91.7/91.9 | 92.2/89.7 | 90.8 | 94.5/94.0 | 92.3 | - |

SOTA PLMs with a similar model size (i.e., BERT, RoBERTa, XLNet, ALBERT$_{large}$, Megatron$_{336M}$ and DeBERTa$_{large}$), DeBERTaV3$_{large}$ shows superior performance on all six tasks. We see a big performance jump on RACE (+2.4%) and SWAG (+2.6%), which require the models to have non-trivial reasoning capability and common-sense knowledge (Lai et al., 2017; Zellers et al., 2018). We conjecture those improvements indicate DeBERTaV3$_{large}$ has a better capability of reasoning and common sense knowledge. Although it is well proved to improve the model capacity by increasing the number of parameters (Raffel et al., 2020; Fedus et al., 2021), compared with larger models,

---

[2]The hidden dimension of ALBERT$_{xxlarge}$ is 4 times of DeBERTa and the computation cost is about 4 times of DeBERTa.

[3]T5 (Raffel et al., 2020) has more parameters (11B). Raffel et al. (2020) only reported the test results of T5 which are not comparable with models mentioned above.

DeBERTaV3$_{large}$ outperforms ALBERT$_{xxlarge}$ and Megatron$_{1.3B}$ by a large margin on all three tasks, as well as outperform Megatron$_{3.3B}$ on both MNLI and SQuAD v2.0. Compared with DeBERTa$_{1.5B}$, which used to be the SOTA NLU models on GLUE and SuperGLUE leaderboards, DeBERTaV3$_{large}$ is still on par with it on MNLI but outperforms it on SWAG.

### 4.1.2 PERFORMANCE ON BASE AND SMALLER MODELS

We evaluate DeBERTaV3$_{base}$, DeBERTaV3$_{small}$, and DeBERTaV3$_{xsmall}$ on two representative tasks, i.e., MNLI and SQuAD v2.0, and summarize the results in Table 5. DeBERTaV3$_{base}$ consistently outperforms DeBERTa$_{base}$ and ELECTRA$_{base}$ by a larger margin than that in the Large models. For example, on MNLI-m, DeBERTaV3$_{base}$ obtains an improvement of +1.8(90.6% vs. 88.8%) over both DeBERTa$_{base}$ and ELECTRA$_{base}$. On SQuAD v2.0 in terms of the EM score, DeBERTaV3$_{base}$ achieves an improvement of +4.9% (85.4% vs. 80.5%) over ELECTRA$_{base}$ and +2.3% (85.4% vs 83.1%) over DeBERTa$_{base}$.

Table 5: Results on MNLI in/out-domain (m/mm) and SQuAD v2.0 development set. TinyBERT$_{small}$(Jiao et al., 2019), MiniLMv2$_{small}$ and MiniLMv2$_{xsmall}$ models are pre-trained with knowledge distillation while BERT$_{small}$, DeBERTaV3$_{small}$ and DeBERTaV3$_{xsmall}$ are trained from scratch with MLM and RTD objective, respectively.

| Model | Vocabulary Size(K) | Backbone #Params(M) | MNLI-m/mm ACC | SQuAD v2.0 F1/EM |
|---|---|---|---|---|
| Base models:12 layers,768 hidden size,12 heads | | | | |
| BERT$_{base}$ | 30 | 86 | 84.3/84.7 | 76.3/73.7 |
| RoBERTa$_{base}$ | 50 | 86 | 87.6/- | 83.7/80.5 |
| XLNet$_{base}$ | 32 | 92 | 86.8/- | -/80.2 |
| ELECTRA$_{base}$ | 30 | 86 | 88.8/- | -/80.5 |
| DeBERTa$_{base}$ | 50 | 100 | 88.8/88.5 | 86.2/83.1 |
| DeBERTaV3$_{base}$ | 128 | 86 | **90.6/90.7** | **88.4/85.4** |
| Small models:6 layers,768 hidden size,12 heads | | | | |
| TinyBERT$_{small}$ | 30 | 44 | 84.5/- | 77.7/- |
| MiniLMv2$_{small}$ | 30 | 44 | 87.0/- | 81.6/ |
| BERT$_{small}$ | 30 | 44 | 81.8/- | 73.2/- |
| DeBERTaV3$_{small}$ | 128 | 44 | **88.2/87.9** | **82.9/80.4** |
| XSmall models:12 layers,384 hidden size,6 heads | | | | |
| MiniLMv2$_{xsmall}$ | 30 | 22 | 86.9/- | 82.3/ |
| DeBERTaV3$_{xsmall}$ | 128 | 22 | **88.1/88.3** | **84.8/82.0** |

Compared with smaller size models with similar model structures, DeBERTaV3$_{small}$ outperforms BERT$_{small}$ (Wang et al., 2020b) by a large margin on those two tasks (i.e., a 6.4% improvement on MNLI-m and a 9.7% F1 score improvement on SQuAD v2.0). Surprisingly, even though DeBERTaV3$_{xsmall}$ has only half the parameters of DeBERTaV3$_{small}$, it performs on par or even better than DeBERTaV3$_{small}$ on these two tasks. We conjecture that this is due to DeBERTaV3$_{xsmall}$ has deeper layers which allows to extract better semantic features. Without knowledge distillation, DeBERTaV3$_{small}$ outperforms MiniLMv2$_{small}$ (Wang et al., 2020a) by 1.2% and 1.3% on MNLI-m and SQuAD v2.0, respectively. DeBERTaV3$_{xsmall}$ outperforms MiniLMv2$_{xsmall}$ (Wang et al., 2020a) by 1.2% and 2.5% on MNLI-m and SQuAD v2.0, respectively. It's worth noting that, even though DeBERTaV3$_{xsmall}$ has only 1/4 backbone parameters of RoBERTa$_{base}$ and XLNet$_{base}$, the former significantly outperforms both models on these two representative tasks (i.e., 0.5% improvement on MNLI-m and 1.5% EM score improvement on SQuAD v2.0). This further demonstrates the efficiency of the DeBERTaV3 models.

### 4.2 MULTILINGUAL MODEL

As an important extension, we extend DeBERTaV3 to multi-lingual. We train the multi-lingual model with the 2.5T CC100 multi-lingual dataset which is the same as XLM-R. We denote the model as mDeBERTaV3$_{base}$. We use the same SentencePiece vocabulary as mT5 which has 250k tokens. The model structure is the same as our base model, i.e., 768 hidden size, 12 layers, and 12 attention heads. It is worth noting that, unlike XLM or XLM-E, we have not trained our model with any parallel data

[4]. The pre-training settings are similar to XLM-R, except that we only train the model with 500k steps instead of 1.5M steps.

Table 6: Results on XNLI test set under the cross-lingual transfer and the translate-train-all settings.

| Model | en | fr | es | de | el | bg | ru | tr | ar | vi | th | zh | hi | sw | ur | Avg |
|---|---|---|---|---|---|---|---|---|---|---|---|---|---|---|---|---|
| Cross-lingual transfer | | | | | | | | | | | | | | | | |
| XLM | 83.2 | 76.7 | 77.7 | 74.0 | 72.7 | 74.1 | 72.7 | 68.7 | 68.6 | 72.9 | 68.9 | 72.5 | 65.6 | 58.2 | 62.4 | 70.7 |
| mT5$_{base}$ | 84.7 | 79.1 | 80.3 | 77.4 | 77.1 | 78.6 | 77.1 | 72.8 | 73.3 | 74.2 | 73.2 | 74.1 | 70.8 | 69.4 | 68.3 | 75.4 |
| XLM-R$_{base}$ | 85.8 | 79.7 | 80.7 | 78.7 | 77.5 | 79.6 | 78.1 | 74.2 | 73.8 | 76.5 | 74.6 | 76.7 | 72.4 | 66.5 | 68.3 | 76.2 |
| mDeBERTaV3$_{base}$ | **88.2** | **82.6** | **84.4** | **82.7** | **82.3** | **82.4** | **80.8** | **79.5** | **78.5** | **78.1** | **76.4** | **79.5** | **75.9** | **73.9** | **72.4** | **79.8** |
| Translate train all | | | | | | | | | | | | | | | | |
| XLM | 84.5 | 80.1 | 81.3 | 79.3 | 78.6 | 79.4 | 77.5 | 75.2 | 75.6 | 78.3 | 75.7 | 78.3 | 72.1 | 69.2 | 67.7 | 76.9 |
| mT5$_{base}$ | 82.0 | 77.9 | 79.1 | 77.7 | 78.1 | 78.5 | 76.5 | 74.8 | 74.4 | 74.5 | 75.0 | 76.0 | 72.2 | 71.5 | 70.4 | 75.9 |
| XLM-R$_{base}$ | 85.4 | 81.4 | 82.2 | 80.3 | 80.4 | 81.3 | 79.7 | 78.6 | 77.3 | 79.7 | 77.9 | 80.2 | 76.1 | 73.1 | 73.0 | 79.1 |
| mDeBERTaV3$_{base}$ | **88.9** | **84.4** | **85.3** | **84.8** | **84.0** | **84.5** | **83.2** | **82.0** | **81.6** | **82.0** | **79.8** | **82.6** | **79.3** | **77.3** | **73.6** | **82.2** |

As XNLI is one of the major benchmarks to measure multi-lingual model generalization performance, we evaluate the performance of mDeBERTaV3 on XNLI across 15 languages. Following previous multi-lingual PLMs, we report both the zero-shot cross-lingual transfer performance and the translate-train-all performance. Zero-shot cross-lingual transfer is to fine-tune the model with English data only and evaluate it on multi-lingual test sets. translate-train-all is to fine-tune the model with English data and multi-lingual data translated from English data which is provided together with the XNLI dataset(Conneau et al., 2018), and then evaluate the model on multi-lingual test sets. As shown in Table 6, mDeBERTaV3$_{base}$ significantly outperforms previous SOTA model XLM-R$_{base}$ on all languages under both settings. With regards to the average score, mDeBERTaV3$_{base}$ obtains an improvement of +3.6% (79.8% v.s. 76.2%) compared with XLM-R$_{base}$ in the cross-lingual transfer setting, as well as achieves an improvement of +3.1% (82.2% v.s. 79.1%) compared with XLM-R$_{base}$ under the translate-train-all setting. These results clearly show the effectiveness of DeBERTaV3 and the disentanglement is simultaneously valuable to multi-lingual pre-training.

All this clearly demonstrates the efficiency of the DeBERTaV3 models. The consistent improvements over a large range of the downstream tasks also show the huge value of improving pre-trained language models.

## 5 Conclusions

In this paper, we propose a novel pre-training paradigm for language models based on the combination of DeBERTa and ELECTRA, two state-of-the-art models that use relative position encoding and replaced token detection (RTD) respectively. We show that simply combining these two models leads to pre-training instability and inefficiency, due to a critical interference issue between the generator and the discriminator in the RTD framework which is well known as the "tug-of-war" dynamics. To address this issue, we introduce a novel embedding sharing paradigm called GDES, which is the main innovation and contribution of this work. GDES allows the discriminator to leverage the semantic information encoded in the generator's embedding layer without interfering with the generator's gradients and thus improves the pre-training efficiency. GDES defines a new way of sharing information between the generator and the discriminator in the RTD framework, which can be easily applied to other RTD-based language models. We conduct extensive analysis and experiments to compare GDES with other alternatives to verify its effectiveness.

Furthermore, we show that DeBERTaV3 with GDES achieves significant improvements over previous state-of-the-art (SOTA) models on various NLU tasks that cover different aspects of natural language understanding. For example, DeBERTaV3$_{Large}$ surpasses other models with a similar architecture by more than 1.37% on the GLUE average score and mDeBERTaV3$_{base}$ beats XLM-R$_{base}$ by 3.6% on the cross lingual transfer accuracy of the XNLI task. These results highlight the effectiveness of all the DeBERTaV3 models and establish DeBERTaV3 as the new SOTA pre-trained language models (PLMs) for natural language understanding at different model scales, i.e., Large, Base, Small and XSmall. Meanwhile, this work clearly shows huge potential to further improve model's parameter efficiency and provide some direction for future studies of far more parameter-efficient pre-trained language models.

---

[4]we expect to have a better model by continually pre-training on parallel data and will release a new model when available

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

# A APPENDIX

## A.1 DATASET

Table 7: Summary information of the NLP application benchmarks.

| Corpus | Task | #Train | #Dev | #Test | #Label | Metrics |
|--------|------|--------|------|-------|--------|---------|
| **General Language Understanding Evaluation (GLUE)** | | | | | | |
| CoLA | Acceptability | 8.5k | 1k | 1k | 2 | Matthews corr |
| SST | Sentiment | 67k | 872 | 1.8k | 2 | Accuracy |
| MNLI | NLI | 393k | 20k | 20k | 3 | Accuracy |
| RTE | NLI | 2.5k | 276 | 3k | 2 | Accuracy |
| WNLI | NLI | 634 | 71 | 146 | 2 | Accuracy |
| QQP | Paraphrase | 364k | 40k | 391k | 2 | Accuracy/F1 |
| MRPC | Paraphrase | 3.7k | 408 | 1.7k | 2 | Accuracy/F1 |
| QNLI | QA/NLI | 108k | 5.7k | 5.7k | 2 | Accuracy |
| STS-B | Similarity | 7k | 1.5k | 1.4k | 1 | Pearson/Spearman corr |
| **Question Answering** | | | | | | |
| SQuAD v1.1 | MRC | 87.6k | 10.5k | 9.5k | - | Exact Match (EM)/F1 |
| SQuAD v2.0 | MRC | 130.3k | 11.9k | 8.9k | - | Exact Match (EM)/F1 |
| ReCoRD | MRC | 101k | 10k | 10k | - | Exact Match (EM)/F1 |
| RACE | MRC | 87,866 | 4,887 | 4,934 | 4 | Accuracy |
| SWAG | Multiple choice | 73.5k | 20k | 20k | 4 | Accuracy |
| **Token Classification** | | | | | | |
| CoNLL 2003 | NER | 14,987 | 3,466 | 3,684 | 8 | F1 |
| **Multi-lingual Natural Language Inference(XNLI)** | | | | | | |
| XNLI$_{cross-lingual}$ | NLI | 393k | 37k | 75k | 3 | Accuracy |
| XNLI$_{translate\ train}$ | NLI | 5.9M | 37k | 75k | 3 | Accuracy |

• **GLUE**. The General Language Understanding Evaluation (GLUE) benchmark is a collection of nine natural language understanding (NLU) tasks. As shown in Table 7, it includes question answering (Rajpurkar et al., 2016), linguistic acceptability (Warstadt et al., 2018), sentiment analysis (Socher et al., 2013), text similarity (Cer et al., 2017), paraphrase detection (Dolan & Brockett, 2005), and natural language inference (NLI) (Dagan et al., 2006; Bar-Haim et al., 2006; Giampiccolo et al., 2007; Bentivogli et al., 2009; Levesque et al., 2012; Williams et al., 2018). The diversity of the tasks makes GLUE very suitable for evaluating the generalization and robustness of NLU models.

• **RACE** is a large-scale machine reading comprehension dataset collected from English examinations in China designed for middle school and high school students (Lai et al., 2017).

• **SQuAD v1.1/v2.0** is the Stanford Question Answering Dataset (SQuAD) v1.1 and v2.0 (Rajpurkar et al., 2016; 2018), two popular machine reading comprehension benchmarks from approximately 500 Wikipedia articles with questions and answers obtained by crowdsourcing. The SQuAD v2.0 dataset includes unanswerable questions about the same paragraphs.

• **SWAG** is a large-scale adversarial dataset for the task of grounded commonsense inference, which unifies natural language inference and physically grounded reasoning (Zellers et al., 2018). SWAG consists of 113k multiple choice questions about grounded situations.

• **CoNLL 2003** (Sang & De Meulder, 2003) is an English dataset consisting of text from a wide variety of sources. It has 4 types of named entities.

• **XNLI** (Conneau et al., 2018) comes with ground truth dev and test sets in 15 languages, and a ground-truth English training set which is same as MNLI training set. The training set has been machine-translated to the remaining 14 languages, providing synthetic training data for these languages as well.

## A.2 PRE-TRAINING DATASET

For DeBERTaV3 pre-training, we use same data as RoBERTa and DeBERTa$_{1.5B}$, which is a combination of Wikipedia, Bookcorpus, CCNews, Stories and OpenWebText. The multi-lingual version

of DeBERTaV3 is trained with 2.5TB CC100 data which is the same as XLM-R. For pre-training, we also sample 5% of the training data as the validation set to monitor the training process. Table 8 compares datasets used in different pre-trained models.

Table 8: Comparison of the pre-training data.

| Model | Wiki+Book 16GB | OpenWebText 38GB | Stories 31GB | CC-News 76GB | Giga5 16GB | ClueWeb 19GB | Common Crawl 110GB | CC100 2.5TB |
|---|---|---|---|---|---|---|---|---|
| BERT | ✓ | | | | | | | |
| XLNet | ✓ | | | | ✓ | ✓ | ✓ | |
| ELECTRA | ✓ | | | | ✓ | ✓ | ✓ | |
| RoBERTa | ✓ | ✓ | ✓ | ✓ | | | | |
| DeBERTa | ✓ | ✓ | ✓ | | | | | |
| DeBERTa$_{1.5B}$ | ✓ | ✓ | ✓ | ✓ | | | | |
| DeBERTaV3 | ✓ | ✓ | ✓ | ✓ | | | | |
| mDeBERTaV3$_{base}$ | | | | | | | | ✓ |

### A.3 GENERALITY OF GDES

To demonstrate the generality of GDES as an enhancement for RTD, we applied it to the ELECTRA setting, which we reproduced with a base model of 12 layers for the discriminator and 6 layers for the generator, both with the same hidden size. We used wikipedia+bookcorpus data to pre-train the model from scratch with a learning rate of 5e-4 and a batch size of 2k for 125k steps. We then evaluated the pre-trained models on MNLI and SQuAD v2.0 using the same setting as in Table 2. The results in Table 9, show that GDES can also improve the training efficiency of RTD in this setting, consistent with the findings in Table 2.

Table 9: Fine-tuning results on MNLI and SQuAD v2.0 tasks of base ELECTRA models trained with different embedding sharing methods.

| Model | MNLI-m/mm Acc | SQuAD v2.0 F1/EM |
|---|---|---|
| ELECTRA$_{base}$ | 85.8/- | -/- |
| ELECTRA$_{base}$ | | |
| ① Reimplemented (ES) | 87.9/87.4 | 85.0/82.3 |
| ② NES | 86.3/85.6 | 81.7/78.9 |
| ③ GDES | **88.3/87.8** | **85.9/83.1** |

### A.4 IMPLEMENTATION DETAILS

Our pre-training almost follows the same setting as DeBERTa (He et al., 2020). The generators are trained with MLM where we randomly replace 15% input tokens with `[MASK]` tokens. The discriminator is trained with RTD which is the same as ELECTRA. The experiments in Section 3 are trained using Wikipedia English data and Bookcorpus data with a batch size of 2k for 125k steps. The experiments in Section 4 are trained using data listed in Table 8 with a batch size of 8k for 500k steps. We list the detailed hyper parameters of pre-training in Table 10. For pre-training, we use Adam (Kingma & Ba, 2014) as the optimizer with weight decay (Loshchilov & Hutter, 2018). For fine-tuning, we use Adam (Kingma & Ba, 2014) as the optimizer for a fair comparison. For fine-tuning, we train each task with a hyper-parameter search procedure, each run taking about 1-2 hours on a DGX-2 node. All the hyper-parameters are presented in Table 11. The model selection is based on the performance on the task-specific development sets.

Our code is implemented based on DeBERTa (He et al., 2020)[5] and ELECTRA (Clark et al., 2020)[6].

---

[5]https://github.com/microsoft/DeBERTa
[6]https://github.com/google-research/electra

Table 10: Hyper-parameters for pre-training DeBERTaV3.

| Hyper-parameter | DeBERTaV3$_{large}$ | DeBERTaV3$_{base}$ | DeBERTaV3$_{small}$ | mDeBERTaV3$_{base}$ | DeBERTaV3$_{base-analysis}$ |
|---|---|---|---|---|---|
| Number of Layers | 24 | 12 | 6 | 12 | 12 |
| Hidden size | 1024 | 768 | 768 | 768 | 768 |
| FNN inner hidden size | 4096 | 3072 | 3072 | 3072 | 3072 |
| Attention Heads | 12 | 12 | 12 | 12 | 12 |
| Attention Head size | 64 | 64 | | 64 | 64 |
| Dropout | 0.1 | 0.1 | 0.1 | 0.1 | 0.1 |
| Warmup Steps | 10k | 10k | 10k | 10k | 10k |
| Learning Rates | 3e-4 | 6e-4 | 6e-4 | 6e-4 | 5e-4 |
| Batch Size | 8k | 8k | 8k | 8k | 2k |
| Weight Decay | 0.01 | 0.01 | 0.01 | 0.01 | 0.01 |
| Max Steps | 500k | 500k | 500k | 500k | 125k |
| Learning Rate Decay | Linear | Linear | Linear | Linear | Linear |
| Adam $\epsilon$ | 1e-6 | 1e-6 | 1e-6 | 1e-6 | 1e-6 |
| Adam $\beta_1$ | 0.9 | 0.9 | 0.9 | 0.9 | 0.9 |
| Adam $\beta_2$ | 0.98 | 0.98 | 0.98 | 0.98 | 0.999 |
| Gradient Clipping | 1.0 | 1.0 | 1.0 | 1.0 | 1.0 |

Table 11: Hyper-parameters for fine-tuning DeBERTaV3 on down-streaming tasks.

| Hyper-parameter | DeBERTaV3$_{large}$ | DeBERTaV3$_{base}$ | DeBERTaV3$_{small}$ | mDeBERTaV3$_{base}$ |
|---|---|---|---|---|
| Dropout of task layer | {0,0.15,0.3} | {0,0.1,0.15} | {0,0.1,0.15} | {0,0.1,0.15} |
| Warmup Steps | {50,100,500,1000} | {50,100,500,1000} | {50,100,500,1000} | {50,100,500,1000} |
| Learning Rates | {5e-6, 8e-6, 9e-6, 1e-5} | {1.5e-5,2e-5, 2.5e-5, 3e-5} | {1.5e-5,2e-5, 3e-5, 4e-5} | {1.5e-5,2e-5, 2.5e-5, 3e-5} |
| Batch Size | {16,32,64} | {16,32,48,64} | {16,32,48,64} | {16,32,48,64} |
| Weight Decay | 0.01 | 0.01 | 0.01 | 0.01 |
| Maximun Training Epochs | 10 | 10 | 10 | 10 |
| Learning Rate Decay | Linear | Linear | Linear | Linear |
| Adam $\epsilon$ | 1e-6 | 1e-6 | 1e-6 | 1e-6 |
| Adam $\beta_1$ | 0.9 | 0.9 | 0.9 | 0.9 |
| Adam $\beta_2$ | 0.999 | 0.999 | 0.999 | 0.999 |
| Gradient Clipping | 1.0 | 1.0 | 1.0 | 1.0 |

