# OpenReview forum: "DeBERTaV3: Improving DeBERTa using ELECTRA-Style Pre-Training with Gradient-Disentangled Embedding Sharing"
_ICLR.cc/2023/Conference — ICLR 2023 poster_

### Official Review · Reviewer_62yM · 2022-10-18

**Confidence:** 5
**Correctness:** 4
**Technical Novelty And Significance:** 2
**Empirical Novelty And Significance:** 3
**Recommendation:** 6

**Clarity, Quality, Novelty And Reproducibility:**

* Clarity and reproducibility should not be an issue
* Quality-wise the paper is well-executed.
* I found the GDES contribution insightful. I could see criticism about it being the only technical contribution but that was not too much of an issue for me.

**Strength And Weaknesses:**

Pros:
- The embedding sharing issue and its fix is discussed clearly and well supported by the author's experiments.
- The results of "NewModel" are empirically quite strong.
- The paper's writing and structure is quite solid. See below for minor typos/comments.
- The authors have done a good job at providing experimental details.

Cons:
- The name "NewModel" across the paper is really not ideal. There are three possibilities: 1) The authors really do intend to adopt this name 2) a macro failed 3) A name was not found in time for the deadline and the authors feared that the DeBERTaV3 name would be more likely to raise comments about the lack of novelty than "NewModel". The "NewModel" will then be replaced upon acceptance/finding a new name. I am not very comfortable with the latter as I feel like it games the system. I will give the benefits of the doubt in this instance.
- I am not a fan of the comparison to larger Megatron models. While it is a larger model, its performance is abnormally low for a model of this size. I think the other comparisons are enough to make your performance clear.
- Minor: Can you confirm that the step time for GDES is only marginally different than ES? It would be great to have this in the paper as further context for Fig 2.


Minor writing issues:
* where the generator’s and the discriminator’s objectives *interference* with each other and slow down the training convergence -> interfere
* section 4.1 first sentence: you likely mean GDES not DES. Also I would spell out DA as disentangled attention as all readers might not remember what it stands. Alternatively, use the abbreviation in Section 2.2.
* page 1 remove space "(Clark et al., 2020) ."


**Summary Of The Paper:**

This paper shows that embedding sharing in ELECTRA hurts the training dynamics and proposes a gradient-disentangled embedding sharing method. This method achieves better results in downstream tasks than both embedding sharing and separate embeddings. The paper then uses this ELECTRA objective on a DeBERTa model architecture, showing best in class performance for models of similar size and often surpassing bigger models across many NLU tasks. They also show that this outperformance extends to smaller model settings and to multilingual models.

The main technical contribution of the paper is in identifying the ELECTRA gradient-sharing issue and in proposing an effective scheme to replace it. While the tests of the combination of ELECTRA and DeBERTa done in the rest of the paper presents less novelty, the high performance of these models for these size means they could be useful to the wider community and so this empirical contribution should not be disregarded.


**Summary Of The Review:**

The GDES contribution is useful and the final model in this paper achieves very good size/performance tradeoffs which will be useful for the wider community. Despite limited novelty and some minor issues highlighted in the Cons, I think it is well executed and slightly above the threshold.

---

> ### Author Response · Authors · 2022-11-16
> **Response**
>
> We appreciate your constructive feedback and recommendations! We will address all the writing issues in our revised version. We will rename our model to DeBERTaV3 to reflect the improvements and facilitate the comparison. The step time of GDES and ES is identical and we will clarify this in our revised version.

---

> > ### Comment · Reviewer_62yM · 2022-11-19
> > **Thank you for the response, maintaining my grade.**
> >
> > Thank you for the response. I believe the current grade still stands after revisions.

---

### Official Review · Reviewer_9Bb4 · 2022-10-22

**Confidence:** 4
**Correctness:** 4
**Technical Novelty And Significance:** 3
**Empirical Novelty And Significance:** 3
**Recommendation:** 8

**Clarity, Quality, Novelty And Reproducibility:**

The paper is very well written, easy to follow, and the argument/evidence/experiment results are solid.

**Strength And Weaknesses:**

The strength of the paper is the following:
1) The goal of the paper is clear. The authors wanted to improve over existing DeBERTa model, with new training approach.
2) The idea of "gradient-disentangle" is solid and novel. The author drew the conclusion based on 3 evidences: the average cosine similarity of token embeddings, the convergence of MLM loss, and the downstream task comparison.
3) The experiment results demonstrating the effectiveness on NLU tasks are comprehensive and solid.

The weakness of the paper:
1) The name "NewModel" could be improved -- this name appears to have nothing to do the DeBERTa, and does not accommodate model backbone at all, so it will be confusing to keep track of.

**Summary Of The Paper:**

The paper trained a new DeBERTa model (named NewModel) which incorporates two additional ingredients: 1) The ELECTRA-style training; 2) The token embedding sharing mechanisms, where the authors proposed gradient-disentangled embedding sharing. The author empirically demonstrated that the existing token sharing is not efficient and hurts model performance, and then show that by not propagating RTD loss to generator, the training can be much better. 3)The author also pre-trained a multi-lingual model and observe larger improvement over strong baselines compared to English models.

**Summary Of The Review:**

I recommend acceptance of this paper, for the following reasons:
1) The gradient-disentangled embedding sharing idea is novel and well supported by three aspects of evidences.
2) The improvement of the model performance is solid and significant, both from the English model and multilingual model.
3) The paper has excellent quality and clarity.
4) While ELECTRA is a known idea, the paper does not merely using the method as it is.

---

> ### Author Response · Authors · 2022-11-16
> **Response**
>
> We appreciate your constructive feedback and recommendations! We will use DeBERTaV3 as the model name.

---

> > ### Comment · Reviewer_9Bb4 · 2022-11-21
> > **Thanks for the clarifications.**
> >
> > The current rating stands as it is.

---

### Official Review · Reviewer_2Wnk · 2022-10-24

**Confidence:** 3
**Correctness:** 3
**Technical Novelty And Significance:** 3
**Empirical Novelty And Significance:** 3
**Recommendation:** 6

**Clarity, Quality, Novelty And Reproducibility:**

This paper is well-written, with adequate training details for reproducibility.
This paper presents a novel and effective embedding-sharing strategy for the DeBERTa + RTD setting.

**Strength And Weaknesses:**

Strengths:
1. A new pre-trained language model is proposed, which can combine the merits of DeBERTa and ELECTRA by enhancing DeBERTa with the RTD pre-training task and an extra GDES strategy to improve the vanilla embedding-sharing strategy from ELECTRA.
2. A few strong pre-trained backbone alternatives will be released that can achieve very competitive performance across different model sizes, downstream tasks, and settings.

Weaknesses:
1. I am not convinced that the newly introduced GDES can improve the training efficiency of the pre-trained models. It has been well-proven that RTD is a sample-efficient pre-training task with thorough comparisons of pre-train FLOPs to other pre-trained language models. However, this paper only utilizes the MLM loss curve of the generator to demonstrate the efficiency of the whole pre-training procedure of both the generator and the discriminator. In real applications, we mainly utilize the larger discriminator for downstream task fine-tuning.
2. The technical contribution is limited. RTD is borrowed from ELECTRA, and I am not sure whether the GDES strategy can generalize well to other MLM + RTD setting (e.g., ELECTRA), except for DeBERTa + RTD.

**Summary Of The Paper:**

This paper proposes to enhance the DeBERTa model with the more sample-efficient replaced token detection (RTD) pre-training task.
To further improve the training efficiency and performance of the vanilla embedding-sharing strategy (ELECTRA) for the DeBERTa + RTD setting, a new gradient-entangled embedding-sharing (GDES) alternative is leveraged. Experiments across different model sizes and downstream tasks demonstrate the effectiveness of the combination of DeBERTa and RTD and the newly introduced GDES strategy.

**Summary Of The Review:**

The newly proposed pre-trained language model achieves very competitive performance with the existing RTD pre-training task and a newly presented embedding-sharing strategy.

---

> ### Author Response · Authors · 2022-11-16
> **Response**
>
> We appreciate your feedback! The generator's convergence is not only important for discriminator-only applications, but also for two main reasons. First, both models share the embeddings, so the generator's semantic learning benefits the discriminator's performance. Second, a stronger generator produces more ambiguous tokens to replace, which challenges the discriminator to learn better representations. Table 2 shows our experimental evidence for this. We also demonstrate that GDES is a general improvement for RTD-like pre-training tasks by applying it to ELECTRA, which we reproduced with a base model setting (12 layers for discriminator, 6 layers for generator with the same hidden size). We pre-trained the model from scratch on wiki+bookcorpus data with a learning rate of 5e-4 and a batch size of 2k for 125k steps. We used the same setting as Table 2 to evaluate the pre-trained models on MNLI and SQuAD v2.0 tasks. The results below indicate that GDES can enhance the training efficiency of RTD, as in Table 2. We will include these results in our appendix in the revised version.
> |ELECTRA Variants|	MNLI|	SQuAD v2.0|
> |--------------|------------|----------------------|
> |ELECTRA Baseline(ES)	|87.9/87.4	|85.0/82.3|
> | NES|	86.3/85.6	|81.7/78.9|
> | GDES|	88.3/87.8	|85.9/83.1|

---

> > ### Comment · Reviewer_2Wnk · 2022-11-28
> > **Thanks for the explanations and experiments.**
> >
> > The responses have addressed part of my concerns, but I think the rating still holds for the updated manuscript.

---

> ### Author Response · Authors · 2022-11-28
> **Kindly ask for re-evaluating our work**
>
> Dear reviewer,
>
> We have carefully updated the manuscript and provided the response according to your feedback and conducted the additional experiments that you requested. We have also clarified and strengthened our arguments and results in our response. We hope that our revisions have addressed some of your concerns and improved the quality and significance of our work. We kindly ask you to re-evaluate our work and consider updating the rating if you find any of the response satisfactory. We appreciate your time and effort in reviewing our work.

---

### Official Review · Reviewer_7CEr · 2022-10-25

**Confidence:** 4
**Correctness:** 4
**Technical Novelty And Significance:** 2
**Empirical Novelty And Significance:** 2
**Recommendation:** 5

**Clarity, Quality, Novelty And Reproducibility:**



Experimentation is problematic. A fair comparison against DeBERTA and ELECTRA alone should involve training both models, again with the same training data. Authors do not state (at least not clearly) that has been the case.

**Details Of Ethics Concerns:**

no comments

**Strength And Weaknesses:**

Strength
The paper is sound and make sense.
DeBERTa and ELECTRA are both top level PLMs, but with significantly different training regimes so its combination makes sense to lead to further improvements.

Weaknesses
Novelty is weak.
Combining (any) models might the actual contribution here. Picking two SOTA models and combining them is not clear to have enough novelty.
The GDES idea is interesting, but doing a thoughtful ablation study with any pairs of PLMs instead will prove authors point.


**Summary Of The Paper:**

The current research piece introduce combines existing pretrained language models as DeBERTa and ELECTRA into a single model.
In addition, authors introduce GDES, a technique that the shared embeddings from the generator and discriminative models interfere with each other by avoiding propagating the gradients due to the discriminative loss.
The model is evaluated in a series of standard NLU dataset, particularly GLUE benchmark, plus other dataset for specific tasks such as QA, NLI and NER varying models sizes showing the effectiveness of their proposed method.




**Summary Of The Review:**

Despite the jump in performance shown by the experimentation the paper do not provide enough novelty..

---

> ### Author Response · Authors · 2022-11-16
> **Response**
>
> We appreciate your encouraging comments and constructive suggestions on our work. We would like to emphasize that this work is not merely a combination of DeBERTa and Electra, but also a novel contribution of GDES. Pre-training is a crucial and expensive technique that can benefit many downstream tasks. For instance, a large model needs at least 128 A100-80G GPUs for more than 10 days to train, excluding any hyper parameter search or machine failure. However, any advancement in pre-training can bring substantial and consistent improvements to downstream tasks. Many previous works have explored pre-training and established strong baselines like Electra and DeBERTa, which are still among the most popular models in Huggingface. This makes it very challenging to achieve new and significant breakthroughs in pre-training, although we believe any incremental progress will be highly valuable for the community due to its foundational and practical impacts. Regarding this work, we acknowledge that the combination of DeBERTa and Electra is straightforward, but we are the first to propose and demonstrate its effectiveness in both pre-training and downstream tasks. Moreover, we claim that the innovation of GDES is substantial. This is based on a deep analysis of this specific combination and it solves a critical issue that has not been addressed before, as well as demonstrating the great performance. Additionally, as suggested by two reviewers here, we also apply GEDS to Electra only model to show its generality.
>
> As requested, we demonstrate the generality of GDES as an enhancement for RTD by applying it to the ELECTRA setting, which we reproduced with a base model of 12 layers for the discriminator and 6 layers for the generator, both with the same hidden size. We used wikipedia+bookcorpus data to pre-train the model from scratch with a learning rate of 5e-4 and a batch size of 2k for 125k steps. We then evaluated the pre-trained models on MNLI and SQuAD v2.0 using the same setting as in Table 2. The results, which we will include in our revision, show that GDES can also improve the training efficiency of RTD in this setting, consistent with the findings in Table 2.
> |ELECTRA Variants|	MNLI|	SQuAD v2.0|
> |--------------|------------|----------------------|
> |ELECTRA Baseline(ES)	|87.9/87.4	|85.0/82.3|
> | NES|	86.3/85.6	|81.7/78.9|
> | GDES|	88.3/87.8	|85.9/83.1|
>
> As the the pre-training data is not shared by ELECTRA, it's impossible to run exactly the same comparison with ELECTRA. Instead, we used the same data and training settings in our ablation study to make fair comparisons.

---

> > ### Comment · Reviewer_7CEr · 2022-12-12
> > **Not convinced**
> >
> > The authors have not address the main concern of my review. "NewModel" is introduced a brand new model, while is just an incremental version of DeBERTa (a combination of DeBERTa and ELECTRA), with RTD as objective (section 3.1). Both models previously introduced. The GDES is in fact novel, but is presented as a secondary contribution by the authors themselves (Abstract and Section 3.3). Conclusions were in fact updated to enforce the novelty of GDES but to my understanding that makes the whole work inconsistent. (i.e. "To
> > address this issue, we introduce a novel embedding sharing paradigm called GDES, which is the main innovation and contribution of this work.")
> >
> > If the main contribution is GDES it should stated on the first line of the conclusions "In this work we introduced GDES to address xxx"
> >
> > I still thing work is below the acceptance threshold, not because the contributions are not there, but because the issues raised above.

---

> > > ### Author Response · Authors · 2022-12-14
> > > **Clarifying the motivation, contribution, and presentation of our work**
> > >
> > > We appreciate the feedback from the reviewers and we respect their perspectives on how to present our ideas and contributions. However, we would like to clarify some points that we think are important to understand our work and its impact.
> > >
> > > First, we have a strong reason to present the combination of DeBERTa and Electra before GDES, because they are not independent methods but rather sequential steps in our pre-training process. We first tried to combine DeBERTa and Electra to leverage their complementary strengths, but we encountered some inefficiency and instability issues. This motivated us to develop GDES, a novel gradient-based method that can dynamically fuse the two models and improve their performance. Moreover, we latter found that GDES can also be applied to Electra only models and achieve similar results. Therefore, we believe that GDES is a general and effective technique that can be extended to other models and tasks.
> > >
> > > Second, we disagree that the combination of DeBERTa and Electra is trivial or irrelevant to the community. On the contrary, we think that it is a valuable contribution to show that two state-of-the-art models can be integrated and enhanced by our methods. We also demonstrate that our combined model outperforms both DeBERTa and Electra on several downstream tasks, and we share our model and code with the community to facilitate further research and applications. We are glad to see that many works have already built on top of DeBERTa-V3 and that many Kaggle solutions have used it to achieve SoTA results.
> > >
> > > Third, we strive to make our paper clear and comprehensive, and we are open to any suggestions that can improve its quality and readability. If the reviewers find any unclear or misleading statements, we will be happy to revise them accordingly. If the reviewers have any specific recommendations on how to restructure or rewrite our paper, we will be happy to consider them as well. However, we hope that the reviewers can evaluate our work based on its originality, significance, and soundness, and not based on their personal preferences of presentation. We also hope that the reviewers can be open-minded and respectful to different ways of expressing and presenting our work, as we are always willing to learn from different opinions and perspectives.
> > >
> > > Please reconsider it based on our feedback.

---

> ### Author Response · Authors · 2022-11-28
> **Kindly ask for re-evaluating our work**
>
> Dear reviewer,
>
> We have carefully updated the manuscript and provided the response according to your feedback and conducted the additional experiments that you requested. We have also clarified and strengthened our arguments and results in our response. We hope that our revisions have addressed some of your concerns and improved the quality and significance of our work. We kindly ask you to re-evaluate our work and consider updating the rating if you find any of the response satisfactory. We appreciate your time and effort in reviewing our work.

---

> ### Author Response · Authors · 2022-12-11
> **Kindly ask for assessment on our revision**
>
> Dear reviewer,
>
> Would you be able to provide us with your feedback on our revision? We would be grateful for any advice you could provide which would help us to improve our work. We are thankful for your time and consideration.
>
> Thanks!

---

### Author Response · Authors · 2022-11-16
**Response to all**

Thanks all reviewers for your thoughtful comments and constructive suggestions. Your comments and suggestions have helped us to improve the paper. We provided responses and clarifications below for each reviewer.

We updated our paper accordingly with changes highlighted,
1. We changed the model name from NewModel to DeBERTaV3 to reflect the improvements
2. We added GDES generality study with ELECTRA base model in Appendix A 3.
3. We addressed writing issues pointed out by reviewer 4.

---

### Decision · Program_Chairs · 2023-01-20

**Decision:**

Accept: poster

**Justification For Why Not Higher Score:**

The technical contribution here is quite limited, and the paper is mostly a combination of existing ideas.

**Justification For Why Not Lower Score:**

While the ideas here are mostly familiar, the execution is good and some novel contributions were required to allow this work to push the current state of the art.

**Metareview: Summary, Strengths And Weaknesses:**

The submission proposes a new pre-training technique that combines ideas from DeBERTa and Electra. Making these idea work together further required the addition of a novel 'gradient disentangled embedding layer'.

The approach is a straightforward but logical evolution of existing ideas, and achieves very strong results. The authors have done a good job with the response and updated submission, which addresses most of the main concerns.

The main weakness is a somewhat limited technical contribution, however overall this is useful work that pushes the current state-of-the-art.

**Note From Pc:**

if the above contains the word "oral" or "spotlight" please see: "oral" presentation means -> notable-top-5% and "spotlight" means -> notable-top-25%. As stated in our emails, we are disassociating presentation type from AC recommendations

**Summary Of Ac-Reviewer Meeting:**

na